# A Novel Technique of Endoscopic Papillectomy with Hybrid Endoscopic Submucosal Dissection for Ampullary Tumors: A Proof-of-Concept Study (with Video)

**DOI:** 10.3390/jcm9082671

**Published:** 2020-08-18

**Authors:** Naminatsu Takahara, Yosuke Tsuji, Yousuke Nakai, Yukari Suzuki, Akiyuki Inokuma, Sachiko Kanai, Kensaku Noguchi, Tatsuya Sato, Ryunosuke Hakuta, Kazunaga Ishigaki, Kei Saito, Yoshiki Sakaguchi, Tomotaka Saito, Tsuyoshi Hamada, Suguru Mizuno, Hirofumi Kogure, Kazuhiko Koike

**Affiliations:** 1Department of Gastroenterology, Graduate School of Medicine, The University of Tokyo, 7-3-1, Hongo, Bunkyo-ku, Tokyo 113-8655, Japan; naminatsu-takahara@umin.ac.jp (N.T.); ytsuji-tky@umin.ac.jp (Y.T.); suzukiyu-int@h.u-tokyo.ac.jp (Y.S.); inokumaa-int@h.u-tokyo.ac.jp (A.I.); kanais-int@h.u-tokyo.ac.jp (S.K.); noguchik-int@h.u-tokyo.ac.jp (K.N.); tatsusatou-tky@umin.org (T.S.); ishigakika-int@h.u-tokyo.ac.jp (K.I.); saitoke-int@h.u-tokyo.ac.jp (K.S.); sakaguchiy-int@h.u-tokyo.ac.jp (Y.S.); tomsaito-gi@umin.ac.jp (T.S.); hamada-tky@umin.ac.jp (T.H.); smizuno-tky@umin.ac.jp (S.M.); hkogure-tky@umin.ac.jp (H.K.); kkoike-tky@umin.ac.jp (K.K.); 2Department of Endoscopy and Endoscopic Surgery, Graduate School of Medicine, The University of Tokyo, 7-3-1, Hongo, Bunkyo-ku, Tokyo 113-8655, Japan; hakuta-tky@umin.ac.jp

**Keywords:** ampullary adenoma, endoscopic papillectomy, hybrid ESD

## Abstract

Background: Endoscopic papillectomy (EP) carries a potential risk of procedure-related adverse events and incomplete resection. Since hybrid endoscopic submucosal dissection (ESD) had been established as an alternative option for relatively large and difficult gastrointestinal tumors, we evaluated a novel EP with hybrid ESD (hybrid ESD-EP) for curative safe margin in this proof-of-concept study. Methods: A total of eight cases who underwent hybrid ESD-EP between 2018 and 2020 were identified from our prospectively maintained database. Hybrid ESD-EP involved a (sub)circumferential incision with partial submucosal dissection, and subsequent snare resection of ampullary tumors, which was performed by two endoscopists with expertise in ESD or endoscopic retrograde cholangiopancreatography. Demographic data and clinicopathological outcomes were retrospectively evaluated. Results: En bloc resection was achieved by hybrid ESD-EP in all eight cases, with the median procedure time of 112 (range: 65–170) minutes. The median diameters of the resected specimens and tumors were 18 and 12 mm, respectively. All lateral margins were clear, whereas vertical margin was uncertain in three (38%), resulting in the complete resection rate of 63%. Postoperative bleeding and pancreatitis developed in each one (13%). No tumor recurrence was observed even in those cases with uncertain vertical margin, after a median follow-up of 244 (range, 97–678) days. Conclusions: Hybrid ESD-EP seems to be feasible and promising in ensuring the lateral resection margin. However, further investigations, especially to secure the vertical margin and to shorten the procedure time, should be required.

## 1. Introduction

Ampullary adenomas, which occur sporadically or in the setting of familial adenomatous polyposis, have malignant potential, and therefore complete resection should be considered [1]. At present, endoscopic papillectomy (EP) has been widely accepted as a minimally invasive alternative to surgery [2,3]. However, conventional EP still has several limitations in terms of safety and effectiveness. In addition to the risk of adverse events (AEs) such as bleeding and pancreatitis, a curative resection rate with clear tumor margin is limited to 87.1%, and local recurrence can occur in up to 11.8% of patients [4].

The curative resection rate of EP is lower compared to that of endoscopic submucosal dissection (ESD), the current standard of care for early gastrointestinal cancer. However, ESD for superficial non-ampullary duodenal epithelial tumors is still technically challenging compared to the gastric or colonic ESD due to the difficult scope maneuverability and its increased risk of AEs due to exposure to the bile and pancreatic juice [5]. In addition, a large tumor size is one of the well-known risk factors for procedure-related AEs [6,7]. Therefore, hybrid ESD, which involves a circumferential incision with partial submucosal dissection and subsequent snare resection, was developed to complement the major drawbacks of ESD [8]. Because most of the submucosal dissection can be replaced by snare resection in hybrid ESD, it may decrease the risk of perforation as well as the procedure time. Based on these findings, we conceived novel EP with hybrid ESD (hybrid ESD-EP) for ampullary tumors to maximize the chance of complete resection, especially a lateral margin, without increasing AEs.

In this proof-of-concept study, we aimed to describe technical tips of hybrid ESD-EP and to investigate its clinical outcomes in patients with ampullary tumors.

## 2. Patients and Methods

### 2.1. Patients

Between May 2018 and March 2020, eight consecutive patients who underwent hybrid ESD-EP for an ampullary adenoma at the University of Tokyo Hospital were retrospectively evaluated. The diagnosis of ampullary adenoma was based on pathological findings from biopsy sampling before the hybrid ESD-EP. Endoscopic ultrasound and contrast computed tomography was performed to evaluate tumor extension into the bile and pancreatic duct. The clinical outcomes were retrieved from our prospectively maintained database and the medical records. This study was approved by the ethics committee of the University of Tokyo Hospital.

### 2.2. Endoscopic Papillectomy with Hybrid Endoscopic Submucosal Dissection

All patients were hospitalized and received the endoscopic procedure under conscious sedation with pethidine hydrochloride and midazolam. The hybrid ESD-EP was performed with the cooperation of two experienced endoscopists who have expertise in ESD (Y.T.) and endoscopic retrograde cholangiopancreatography (ERCP) (N.T.), using a gastrointestinal endoscope and duodenoscope (GIF-H290T or PCF-H290T and TJF-260V; Olympus Medical Systems, Tokyo, Japan). We routinely use a soft hood (F-030, TOP Co., Tokyo, Japan) for gastrointestinal endoscope. Video 1 shows a case of hybrid ESD-EP (Figure 1a). After the local injection of sodium hyaluronate solution with indigo-carmine, the anal side of the tumor was first incised, followed by a (sub)circumferential incision using a Dual Knife (KD-650Q; Olympus Medical Systems, Tokyo, Japan) (Figure 1b). Then, trimming was performed deep enough to cut the muscularis mucosa, especially on the lateral and anal sides of the lesion. The submucosal dissection was continued until as large as possible (Figure 1c). Subsequently, with the change in the operator as well as the endoscope, the tumor was grasped along the excision line from the cranial side of the encircling fold to the caudal side of the major papilla, and mucosal resection was performed using a standard loop snare (Captivator; Boston Scientific Japan, Tokyo, Japan) (Figure 1d,e). The settings of the VIO300 D electrical unit (Erbe Elektromedizin, Tübingen, Germany) were “Endo Cut I” (effect 2, duration 2, interval 2) for mucosal incision, “swift coagulation” (effect 3, 45 W) for submucosal dissection, “Endo Cut Q” (effect 2, duration 1, interval 3, Forced Coag, effect3, 30 W) for tumor resection, and ‘Soft Coagulation’ (effect 5, 50 W) for hemostasis.

After the tumor resection, a 7 Fr biliary stent (Flexima, Boston Scientific, Tokyo, Japan) and a 5 Fr pancreatic stent (Geenen; Cook Medical Japan, Tokyo, Japan) were placed without sphincterotomy. Before and after stent placement, electrosurgical hemostatic forceps (HDB2422W; Pentax Medical, Tokyo, Japan) was used on the mucosal vessels of the ulcer bed to control bleeding. Prophylactic coagulation was routinely performed when visible vessels was detected within the mucosal defect. Endoclips (EZ clip; Olympus Medical Systems, Tokyo, Japan and/or Sure Clip; Micro-Tech Co., Ltd., Nanjing, China) were used to attempt to close the mucosal defect to prevent postoperative bleeding and perforation (Figure 1f). The post-procedural ulcer was shielded with polyglycolic acid sheets and fibrin glue at the physicians’ discretion [9].

### 2.3. Histopathological Evaluation

The resected specimens were fixed in 10% buffered formalin. Histopathologic type, lesion size, depth of invasion, tumor involvement in the lateral and vertical margins, and lymphovascular invasion were evaluated.

### 2.4. Follow-Up after the Hybrid ESD-EP

As a rule, oral intake was resumed the day after the procedure in cases without AEs. The biliary and pancreatic stents were left in place for at least one week unless AEs required endoscopic interventions, for example, if bleeding and pancreatitis developed. At the time of endoscopic stent removal, the papillectomy site was endoscopically evaluated for any signs of residual tumor or bleeding. After the discharge, patients visited the outpatient clinic within one month. Follow-up endoscopy was performed within 6 months and repeated every 12 months even in the absence of recurrence.

### 2.5. Endpoints and Definition

The endpoints of this study included the procedure time, en bloc resection rate, complete resection rate, and AEs. Procedure time was defined as the time from starting tumor resection to and finish the whole procedure. En bloc resection was defined as the resection of a tumor virtually without division into segments. Complete resection was histologically assessed as an en bloc resection with both lateral and vertical margin free from dysplasia/neoplasia, whereas incomplete resection was considered in which lateral or vertical margins testing positive or uncertain for the excised tumor. AEs and their severity were defined in accordance with the American Society for Gastrointestinal Endoscopy Lexicon classification systems [10]. Bleeding was defined as intra-procedural if it occurred during the procedure and required endoscopic hemostasis, and post-procedural if hematemesis and/or melena, or a hemoglobin drop by >2 g/dL was observed after discharge from the endoscopy room and within 2 weeks. Pancreatitis was defined as abdominal pain with a 3-fold increase in serum amylase/lipase 24 h to 2 weeks after the procedure.

The final analysis was based on the follow-up information, which was received until May 2020.

## 3. Results

### 3.1. Patients’ Characteristics

The baseline characteristics and procedure outcomes are shown in Table 1. The median age was 63 years, and five patients were male. One patient had a diagnosis of familial adenomatous polyposis. Clopidogrel was discontinued prior to the procedure in one patient with cardiovascular disease; otherwise, no other patients were on antithrombotic agents.

### 3.2. Procedures’ Outcomes and Pathological Findings

All of the ampullary lesions were successfully resected en bloc by the hybrid ESD-EP with the median procedure time of 112 (range; 65–170) minutes. It took a median of 42 (range, 27–80) minutes to make the (sub)circumferential incision and subsequent partial submucosal dissection. In a patient who had a prior history of sphincterotomy for biliary stone removal, the incision around the tumor resulted in subcircumferential. After tumor resection using the snare, both biliary and pancreatic stents were successfully placed in all patients. No residual tumor was endoscopically detected after hybrid ESD-EP. Hemostasis using hemostatic forceps to control intraprocedural bleeding was required in five (63%). Eventually, the mucosal defect was closed by clips alone in two (25%) and in combination with shielding by polyglycolic acid sheets with fibrin glue in six (75%). Postoperative bleeding (moderate) and pancreatitis (moderate) developed on the next day of the procedure in each one (13%). No perforation or procedure-related mortality was observed.

The median diameters of the resected specimens and tumors were 18 (range, 14–28) and 12 (range, 8–16) mm, respectively. There was no distinct residual tumor in lateral margins, whereas vertical margins were uncertain due to the burning effect in three (38%), resulting in the complete resection rate of 63% (Table 2). All patients were alive without tumor recurrence on the follow up endoscopy, including three patients with an unclear vertical margin, after a median follow-up of 8.0 (range, 3.2–22.3) months.

## 4. Discussion

This preliminary proof-of-concept case series demonstrated the feasibility of novel hybrid ESD-EP, which involved a (sub)circumferential incision with partial submucosal dissection and subsequent snare resection, for ampullary adenoma. En bloc resection and complete resection rates were 100% and 63%, respectively. Of note, a clear lateral margin was achieved in all eight cases as we intended, but vertical margin was uncertain in three cases. Postoperative bleeding (moderate) and pancreatitis (moderate) developed in each one (13%). No tumor recurrence was observed even in those cases with uncertain vertical margin, after a median follow-up of 244 days.

There are several hurdles for complete resection of ampullary adenoma. First of all, the precise endoscopic evaluation of tumor extension is necessary. Recent studies reported that narrow-band imaging showed a higher ability to enhance the tumor margin than that in chromoendoscopy or white light imaging, resulting in successful EP with a clear lateral margin in all cases [11]. However, even if we could evaluate tumor extension using these modalities, it is still technically challenging to catch the tissue with clear margin and resect the tumor en bloc using a snare. In cases with flat extension around the ampulla, the tumor can be easily slipped off from the snare. Indeed, some studies with conventional EP reported that a clear lateral margin was obtained only in 58%–78% [12,13]. In our hybrid ESD-EP, prior (sub)circumferential incision and partial submucosal dissection allow us to grasp the tumor tissue without any risk of incision by the snare into the tumor. Although it is difficult to observe the whole ampullary lesion with a forward-viewing endoscope, the high-resolution image with NBI can recognize the tumor margin and subsequent (sub) circumferential incision, and partial submucosal dissection was successfully achieved with the expertise of ESD [14,15,16,17]. Subsequently, the tumor resection was achieved under direct visualization of a duodenoscope, achieving a clear lateral margin in all cases.

While we successfully confirmed lateral margin by the hybrid ESD-EP technique, vertical margin is still an issue. A randomized controlled trial showed that EP with submucosal injection was associated with a low complete resection rate due to positive vertical margin, compared to conventional EP (50.0% vs. 80.8%, *p* = 0.02) [18]. The authors speculated that submucosal injection fails to lift the center area due to the presence of the bile duct and might to lead to a residual tumor in the vertical margin, although in this randomized controlled trial tumor, persistence at 1 month and recurrence at 1 year were both similar irrespective of submucosal injection [18]. Furthermore, the pathological evaluation of the margins of resected ampullary tumor specimens is often difficult because of the burning effects of EP [19]. A retrospective study reported that nearly a quarter of patients had uncertain tumor margins after EP and 2 out of 15 adenoma cases with positive/uncertain margin developed recurrence after 1.0–5.0 months of EP [12]. However, most of these lesions are endoscopically manageable without surgical interventions [20]. In our series, the vertical margin was uncertain in three cases due to the burning effects but no tumor recurrence was observed after a median follow-up of 244 days. In a pilot randomized controlled trial comparing electrical pulse cut mode, the Autocut mode without coagulation might prevent these burning effects and allow better pathological evaluation [21].

Safety is a major concern to be discussed here. In this study of hybrid ESD-EP, one instance moderate bleeding and pancreatitis was observed but there was no severe morbidity or mortality. Because a systematic review showed that pancreatic stent had a clear benefit in the prevention of EP-related pancreatitis, we routinely put a pancreatic stent in all cases [22]. On the other hand, no effective prophylaxis for bleeding had been established to date. In clinical practice, the closure of post-EP ulcer using clips is now widely performed, although it is often difficult to attach the clip with the duodenoscope [23]. Given the promising efficacy of tissue shielding with polyglycolic acid sheets and fibrin glue for colorectal ESD [9], these devices may decrease EP-related bleeding. There was one report of this combination for management of perforation during EP [24]. However, the risk of obstruction of both biliary and pancreatic orifices should be considered without stent placement.

This study has several limitations. Firstly, this was a preliminary case series involving only two competent endoscopists at a single tertiary referral center. Therefore, there is a concern about generalizability, but the strength of our technique is that the devices used in hybrid ESD-EP are commonly available nowadays. Secondly, the long procedure time should be considered because it may have an increased risk of anesthesia-related AEs such as aspiration pneumonia. The use of traction devices for submucosal dissection may shorten the procedure time [25]. Thirdly, the median follow-up of 8 months might be too short to detect tumor recurrence in cases with low-grade adenoma. Finally, submucosal injection for the treatment of ampullary adenomas may be associated with an insufficient vertical margin. This potential disadvantage of hybrid ESD-EP should be clarified in prospective randomized trials comparing to the conventional EP.

In conclusion, this proof-of-concept study demonstrated that novel hybrid ESD-EP was safe and feasible for endoscopic resection of ampullary adenoma, confirming the clear lateral margin. The technique might be associated with an insufficient vertical margin as well as a long procedure time and further studies are required.

## Figures and Tables

**Figure 1 jcm-09-02671-f001:**
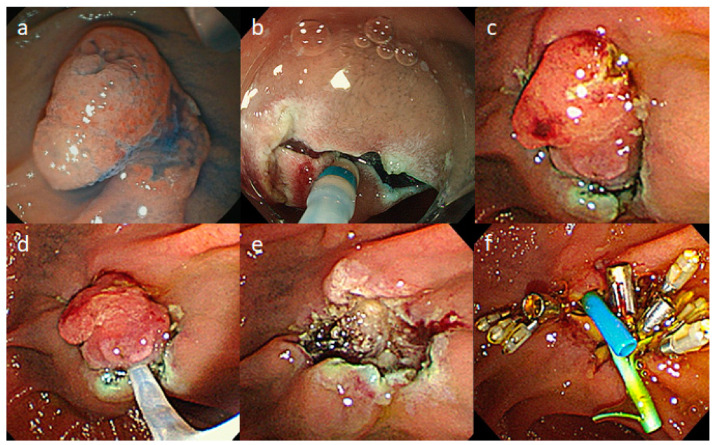
Endoscopic papillectomy with hybrid endoscopic submucosal dissection. Figure legend: (**a**): The ampullary adenoma with the diameter of 20 mm; (**b**): A circumferential incision was made after the local injection of sodium hyaluronate solution; (**c**): The submucosal dissection was continued until as large as possible; (**d**): The tumor was grasped along the excision line; (**e**): The tumor was resected using a standard loop snare; (**f**): Both biliary and pancreatic stents were placed and the ulcer was closed using clips.

**Table 1 jcm-09-02671-t001:** Baseline characteristics and outcomes of the procedures.

Case	Age, Yrs and Sex	Comorbidity	Procedure Time, min	Resection Method	Biliary Stent	Pancreatic Stent	Hemostasis or Prophylaxis of Bleeding	Intra-Procedural Bleeding	Post-Procedure Adverse Events
1	62, F	HTN	85	En bloc	7 Fr 7 cm	5 Fr 7 cm	Coagulation/PGA/fibrin glue	+	-
2	69, F	-	170	En bloc	7 Fr 7 cm	5 Fr 5 cm	Coagulation/PGA/fibrin glue	+	-
3	56, F	-	112	En bloc	7 Fr 7 cm	5 Fr 5 cm	Coagulation/PGA/fibrin glue	+	Bleeding (moderate)
4	49, M	Asthma	65	En bloc	7 Fr 7 cm	5 Fr 7 cm	Clipping	-	Pancreatitis (moderate)
5	63, M	HTN, DM	115	En bloc	7 Fr 7 cm	7 Fr 7 cm	Coagulation/Clipping/PGA/fibrin glue	-	-
6	63, M	HTN, Colon cancer	110	En bloc	7 Fr 7 cm	5 Fr 7 cm	Coagulation/Clipping	+	-
7	77, M	HTN, DM, Old MI, Emphysema	112	En bloc	7 Fr 7 cm	5 Fr 7 cm	Clipping/PGA/fibrin glue	-	-
8	74, M	HTN, FAP	150	En bloc	7 Fr 7 cm	5 Fr 7 cm	Coagulation/Clipping/PGA/fibrin glue	+	-

ASA, American Society of Anesthesiologists; HTN, hypertension; DM, diabetes mellitus; MI, myocardial infarction; FAP, familial adenomatous polyposis; PGA, Polyglycolic acid.

**Table 2 jcm-09-02671-t002:** Pathological findings.

Case	Size of Resected Specimen/Tumor, mm	Pathological Diagnosis	Complete Resection	Lateral Margin	Vertical Margin
1	15/14	Low grade adenoma	+	Negative	Negative
2	18/12	Low grade adenoma	-	Negative	Uncertain
3	23/12	Low grade adenoma	+	Negative	Negative
4	14/12	Low grade adenoma	+	Negative	Negative
5	28/24	Low grade adenoma	-	Negative	Uncertain
6	23/16	Low grade adenoma	+	Negative	Negative
7	18/8	Low grade adenoma	+	Negative	Negative
8	15/12	Low grade adenoma	-	Negative	Uncertain

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
