# Peer review of "A Novel Technique of Endoscopic Papillectomy with Hybrid Endoscopic Submucosal Dissection for Ampullary Tumors: A Proof-of-Concept Study (with Video)"

_jcm, 2020, doi:10.3390/jcm9082671_

Round 1
Reviewer 1 Report
This is an interesting proof-of-concept study reporting on a small cohort of patients undergoing hybrid ESD-Papillectomy (ESD-EP) of adenomas of the papilla of Vater, aiming at decreasing the risk of residual/recurrent adenoma, especially on the lateral margins of resection.The technique, with the limitations of a retrospective study and a small number of patients, seems to be effective (no residual or recurrent adenomas) and therefore merits to be reported. However I have some queries and comments that I would like the Authors to address.
page 2, line 72: which kind of cap was used?
page 2, line 73: was indigo-carmine added to the injection solution?
page 2, line 74: where was the circumferential incision started? The oral side is probably the less important (but also the easiest) to incise because it is usually possible to place the snare over the transverse fold of the infundibulum. How often was the incision "(sub)circumferential" ?
page 2, line 90: how often was it possible to completely close the mucosal defect with clips? Which kind of clips were used?
Table 1: an average time of 2 hours to complete the procedure is quite long. Are there procedural steps that can be modified to speed up the procedure without impairing its efficacy?
Table 2: only 3 out of 8 patients had an adenoma bigger than 2 cm, and all had LGD adenoma. I wonder if this technique finds its best indication only in patients with large adenomas with a lateral spreading tumor appearance, especially if harboring foci of HGD or intramucosal carcinoma. Please add a comment in the discussion.
Author Response
We greatly thank the reviewers for the relevant suggestions, all of which have contributed to a substantial amelioration of our manuscript. We provide here the point-by-point replies to the comments.
<Reviewer 1>
This is an interesting proof-of-concept study reporting on a small cohort of patients undergoing hybrid ESD-Papillectomy (ESD-EP) of adenomas of the papilla of Vater, aiming at decreasing the risk of residual/recurrent adenoma, especially on the lateral margins of resection.The technique, with the limitations of a retrospective study and a small number of patients, seems to be effective (no residual or recurrent adenomas) and therefore merits to be reported. However I have some queries and comments that I would like the Authors to address.
#1. page 2, line 72: which kind of cap was used?
We add a following sentence after page 2, line 72;
“We routinely use a soft hood (F-030, TOP Co., Tokyo, Japan) for gastrointestinal endoscope.”
#2. page 2, line 73: was indigo-carmine added to the injection solution?
#3. page 2, line 74: where was the circumferential incision started? The oral side is probably the less important (but also the easiest) to incise because it is usually possible to place the snare over the transverse fold of the infundibulum. How often was the incision "(sub)circumferential" ?
We modified the sentence in page 2, line 73-74 as follows;
“After the local injection of sodium hyaluronate solution with indigo-carmine, the anal side of the tumor was first incised, followed by a (sub)circumferential incision using a Dual Knife ////.
As the reviewer pointed out, the oral side incision seems less important for successful EP. In our series, only one case intentionally underwent a sub-circumferential incision left the oral side of the tumor as it was because of the submucosal fibrosis due to the prior sphincterotomy (case #2).
We added a following sentence in page 6, line 4;
“In a patient who had a prior history of sphincterotomy for biliary stone removal (case #2), the incision around the tumor resulted in subcircumferential.”
#4. page 2, line 90: how often was it possible to completely close the mucosal defect with clips? Which kind of clips were used?
We routinely use EZ clip (Olympus Medical Systems) and/or Sure Clip (Micro-Tech Co., Ltd, Nanjing, China) at the physicians’ discretion. As shown in page 6, line7-9, the mucosal defect was closed by clips alone in 2 (25%) and in combination with shielding by polyglycolic acid sheets with fibrin glue in 6 (75%).
We modified the sentence in page 2, Line 90 as follows;
“Endoclips (EZ clip; Olympus Medical Systems and/or Sure Clip; Micro-Tech Co., Ltd, Nanjing, China) were used to attempt ///.”
#5. Table 1: an average time of 2 hours to complete the procedure is quite long. Are there procedural steps that can be modified to speed up the procedure without impairing its efficacy?
Our hybrid ESD-EP consists of four steps;1) circumferential incision and submucosal resection, 2) snaring and tumor resection, 3) biliary and pancreatic stent placement, and 4) hemostasis.
Compared to the conventional EP, the first step of circumferential incision and submucosal dissection is an additional step for ESD-EP which can contribute to prolong the procedure time. Recently, several traction devices have been reported to facilitate ESD and shorten procedure time. These methods can complement the procedure limitation.
We add a sentence in page 7, line 70 as follows;
Secondly, the long procedure time should be considered because it may have an increased risk of anesthesia-related AEs such as aspiration pneumonia. Usage of traction devices for submucosal dissection may shorten the procedure time [25].
#6. Table 2: only 3 out of 8 patients had an adenoma bigger than 2 cm, and all had LGD adenoma. I wonder if this technique finds its best indication only in patients with large adenomas with a lateral spreading tumor appearance, especially if harboring foci of HGD or intramucosal carcinoma. Please add a comment in the discussion.
Because this was a proof-of-concept study, we enrolled consecutive cases with ampullary adenoma, irrespective of the tumor size. However, considering the potential advantage of hybrid ESD-EP, it can be a good option for those with large and/or laterally spreading tumor, as the reviewer pointed out. Whereas, there is a still controversy in the indication of EP for ampullary carcinoma. We usually recommend surgery for cases with carcinoma even if it seemed to be intramucosal tumor because of the inherent risk of metastasis.
We add a sentence in Discussion (page 6, Line 35) as follows;
In our hybrid ESD-EP, prior (sub)circumferential incision and partial submucosal dissection allow us to grasp of the tumor tissue without any risk of incision by the snare into the tumor. Thus, we believe this procedure can commit to achieve en bloc resection with clear tumor margin especially in cases with large and/or laterally spreading tumor.
Reviewer 2 Report
The Authors describe a novel endoscopic technique to treat ampullary lesions, and to improve the complete resection rate with snare resection in addition to ESD. The application of this technique for the treatment of ampullary lesions is technically challenging and could represent an alternative approach to the standard EP. In all cases an en bloc resection was performed and lateral margins were negative, whereas vertical margins were uncertain in more than one third of cases.
In the Method section (62-64), you state that diagnosis of adenoma was based on the pre-procedural biopsy sampling, but you should better clarify the indications to the procedure. Could you please specify if all the patients were diagnosed with "adenoma" at biopsy? Did you perform an endoscopic ultrasound to better assess the depth of invasion before the procedure? Please, add these information in the Method section.
Eight patients were included, and after a median follow-up of 8 months (range, 3-23) they showed no tumor recurrence. This study consists in a preliminary case series, as correctly stated (67), since this is a novel technique applied for a relatively rare disease. The reported nihil recurrence rate could be due to the correct selection of cases (i.e. pre-procedural EUS), and to the final histology reported (all "low-grade adenoma" assessed on en bloc resections), but also to the short time of follow-up (median less than 1 year). Since in 3 out of 8 cases the vertical margins resulted uncertain, the time of follow-up represents a limitation of this study for the evaluation of disease recurrence, and please report a statement before the conclusions. Moreover, to avoid any confusion, report the follow-up time in "months" and not in "days".
Author Response
We greatly thank the reviewers for the relevant suggestions, all of which have contributed to a substantial amelioration of our manuscript. We provide here the point-by-point replies to the comments.
<Reviewer 2>
The Authors describe a novel endoscopic technique to treat ampullary lesions, and to improve the complete resection rate with snare resection in addition to ESD. The application of this technique for the treatment of ampullary lesions is technically challenging and could represent an alternative approach to the standard EP. In all cases an en bloc resection was performed and lateral margins were negative, whereas vertical margins were uncertain in more than one third of cases.
#1. In the Method section (62-64), you state that diagnosis of adenoma was based on the pre-procedural biopsy sampling, but you should better clarify the indications to the procedure. Could you please specify if all the patients were diagnosed with "adenoma" at biopsy? Did you perform an endoscopic ultrasound to better assess the depth of invasion before the procedure? Please, add these information in the Method section.
We appreciate this valuable comment.
In addition to the endoscopic observation with tissues sampling, we routinely perform EUS and contrast CT to evaluate local progression as well as distant metastasis for making a treatment strategy for ampullary tumors. EUS has a high ability to detect tumor extension into the bile and pancreatic duct.
#2. Eight patients were included, and after a median follow-up of 8 months (range, 3-23) they showed no tumor recurrence. This study consists in a preliminary case series, as correctly stated (67), since this is a novel technique applied for a relatively rare disease. The reported nihil recurrence rate could be due to the correct selection of cases (i.e. pre-procedural EUS), and to the final histology reported (all "low-grade adenoma" assessed on en bloc resections), but also to the short time of follow-up (median less than 1 year). Since in 3 out of 8 cases the vertical margins resulted uncertain, the time of follow-up represents a limitation of this study for the evaluation of disease recurrence, and please report a statement before the conclusions. Moreover, to avoid any confusion, report the follow-up time in "months" and not in "days".
No tumor recurred even in cases with uncertain vertical margin during the follow-up period. According to the reviewer comment, we state it may be confounded by several factors including patient selection as well as short follow-up duration.
We modify to report the follow-up time in "months (page 6, line 16) and add a limitaion in page 7, line 71 as follows;
“ after a median follow-up of 8.0 (range, 3.2-22.3) months”
“Thirdly, the median follow-up of 8 months might be too short to detect tumor recurrence in cases with low-grade adenoma.”